# PROBABILISTIC SEMANTIC EMBEDDING

## ABSTRACT

We present an extension of a variational auto-encoder that creates semantically rich coupled probabilistic latent representations that capture the semantics of multiple modalities of data. We demonstrate this model through experiments using images and textual descriptors as inputs and images as outputs. Our latent representations are not only capable of driving a decoder to generate novel data, but can also be used directly for annotation or classification. Using the MNIST and Fashion-MNIST datasets we show that the embedding not only provides better reconstruction and classification performance than the current state-of-the-art, but it also allows us to exploit the semantic content of the pretrained word embedding spaces to do tasks such as image generation from labels outside of those seen during training.

## 1 INTRODUCTION

In the last few years one of the most exciting developments in machine learning has been in the development of unsupervised learning models to learn *latent* or *embedding* spaces as exemplified by variational auto-encoders (VAEs) (Kingma & Welling, 2013) for images and word2vec (Mikolov et al., 2013) for words. In this paper we propose a novel framework for coupling together such spaces allowing us to associate an image with a natural language description through a powerful generative probabilistic-semantic embedding in which both images and text are represented as probability distributions in a common latent space.

Our Probabilistic Semantic Embedding model, PSE, is an extension of a VAE, where, rather than use a Kullback-Leibler (KL) divergence term between the image distributions and a single prior, we use a KL divergence between the image distribution and a distribution learned from text. We show this schematically in Figure 1. Both image and text inputs are fed to networks that output the parameters describing normal distributions in the same vector space. Throughout the paper we use images and text as our working examples of the data modalities, although our model is itself generic and could be applied to any modality. Furthermore, although not the focus of this paper, it can also be trivially extended to an arbitrary number of additional input modalities (for example audio).

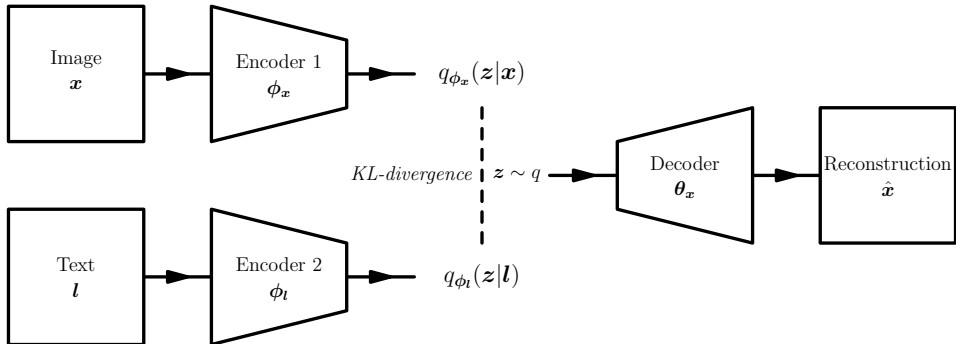

Figure 1: Schematic of the PSE model.

Despite its deceptive simplicity, this model creates a powerful latent space representation where both images and textual descriptors are represented as probability distributions. The geometry of the latent space captures both visual and semantic similarity between classes of object. Unlike current state-of-the-art models such as the Conditional VAE (CVAE), our model is not handicapped by an

assumption of independence between textual terms or labels. We demonstrate that the model allows for modes of operation in which images can be generated directly from labels, or from other images, or both, and, labels predicted from images. We also further demonstrate how, by encoding the text using pre-trained word embedding vectors, we can also exploit the semantics of the word embedding space, allowing us to generate images using words that are outside of the vocabulary of labels used to train our model, whilst at the same time allowing the extended model to produce better classification accuracy on unlabelled images.

We compare our proposed PSE model against a range of state-of-the-art models for conditional and joint generative modelling of images and labels, and demonstrate that our model is capable of lower reconstruction error and higher classification accuracy. We show how prior knowledge from a pre-trained word2vec embedding improves both classification and reconstruction performance.

The remainder of the paper is structured as follows: In section 2 we describe our method in detail. Related work is described in Section 3. In Section 4 we demonstrate our method using one-hot encoded labels and compare with the state-of-the-art. In section 5 we explore how the model can be extended by using pre-trained word embeddings to capture relationships that are not explicit from the labelled image data used to train the model. Conclusions and ideas for future research directions can be found in Section 6.

## 2 PROPOSED METHOD

Our method can be seen as an extension of the standard VAE. Recall that in a VAE we have a data set, $\mathcal{D} = \{\boldsymbol{x}_i\}_{i=1}^N$, that maps an input image $\boldsymbol{x}_i$, through an encoder network with parameters $\boldsymbol{\phi}$, to a distribution, $q_{\boldsymbol{\phi}}(\boldsymbol{z}|\boldsymbol{x}_i)$, in a latent space. During training we sample a vector $\boldsymbol{z}$ from $q_{\boldsymbol{\phi}}(\boldsymbol{z}|\boldsymbol{x}_i)$ that is passed to a decoder network with parameters $\boldsymbol{\theta}$ to generate a reconstruction $\hat{\boldsymbol{x}} = \mathrm{Dec}(\boldsymbol{z}|\boldsymbol{\theta})$. We can use the reconstruction to generate image vectors $\boldsymbol{x}'$ from a distribution $p(\boldsymbol{x}'|\hat{\boldsymbol{x}})$, often taken to be $\mathcal{N}(\boldsymbol{x}'|\hat{\boldsymbol{x}}, \sigma^2\,\boldsymbol{I})$. As derived in the original VAE paper (Kingma & Welling, 2013), we can write down the Evidence Lower BOund (ELBO) as:

$$\mathrm{elbo}(\boldsymbol{x}) = \mathbb{E}_{q_{\boldsymbol{\phi}}(\boldsymbol{z}|\boldsymbol{x})}\big[\log\big(p_{\theta}(\boldsymbol{x}|\boldsymbol{z})\big)\big] - D_{\mathrm{KL}}\big(q_{\boldsymbol{\phi}}(\boldsymbol{z}|\boldsymbol{x})\,\big\|\,p(\boldsymbol{z})\big) \tag{1}$$

where $p_{\theta}(\boldsymbol{x}|\boldsymbol{z}) = p(\boldsymbol{x}'|\hat{\boldsymbol{x}})$ and $p(\boldsymbol{z}) = \mathcal{N}(\boldsymbol{z}; 0, \boldsymbol{I})$ is a "prior" distribution. To train the network we maximise $\mathrm{elbo}(\boldsymbol{x})$ for all $\boldsymbol{x}$ sampled from $\mathcal{D}$. To use the VAE as a generative model we would sample $\boldsymbol{z}$ from the prior $p(\boldsymbol{z})$ and feed this to the decoder.

In the PSE model, we consider a data set made up of image-label pairs $\mathcal{D} = \{(\boldsymbol{x}_i, \boldsymbol{l}_i)\}_{i=1}^N$. The images are given to an encoder that outputs the parameters of a distribution $q_{\boldsymbol{\phi}_x}(\boldsymbol{z}|\boldsymbol{x})$, while the labels are fed to a second encoder network with weights $\boldsymbol{\phi}_l$ that output the parameters for a distribution $q_{\boldsymbol{\phi}_l}(\boldsymbol{z}|\boldsymbol{l})$. In our experiments, we choose the parameters output by both encoder networks to be the means and variances of a multivariate normal distribution with a diagonal covariance matrix, but other reparameterisable distributions are possible. In training we sample a random latent vector $\boldsymbol{z}$ from $q_{\boldsymbol{\phi}_x}(\boldsymbol{z}|\boldsymbol{x})$ and feed it to a decoder network to obtain a reconstruction image $\hat{\boldsymbol{x}} = \mathrm{Dec}(\boldsymbol{z}|\boldsymbol{\theta}_x)$. The only modification to the objective function of a standard VAE is that we replace $p(\boldsymbol{z})$ by $q_{\boldsymbol{\phi}_l}(\boldsymbol{z}|\boldsymbol{l})$. That is, we maximise

$$L_{\mathrm{PSE}}(\boldsymbol{x}, \boldsymbol{l}) = \mathbb{E}_{q_{\boldsymbol{\phi}}(\boldsymbol{z}|\boldsymbol{x})}\big[\log\big(p_{\boldsymbol{\theta}_x}(\boldsymbol{x}|\boldsymbol{z})\big)\big] - D_{\mathrm{KL}}\big(q_{\boldsymbol{\phi}_x}(\boldsymbol{z}|\boldsymbol{x})\,\big\|\,q_{\boldsymbol{\phi}_l}(\boldsymbol{z}|\boldsymbol{l})\big) \tag{2}$$

averaged over all pairs $(\boldsymbol{x}_i, \boldsymbol{l}_i)$ drawn from the new data set.

**Image Generation.** To use the PSE model as an image generator we would put in a label $\boldsymbol{l}$ and sample a latent vector $\boldsymbol{z}$ from $q_{\boldsymbol{\phi}_l}(\boldsymbol{z}|\boldsymbol{l})$. This is given to the decoder to obtain an image $\hat{\boldsymbol{x}} = \mathrm{Dec}(\boldsymbol{z}, \boldsymbol{\theta}_x)$.

**Image Classification.** To use the model for image classification we can put in an image $\boldsymbol{x}$ and encode this to a (Normal) probability distribution $q_{\boldsymbol{\phi}_x}(\boldsymbol{z}|\boldsymbol{x})$. We choose to use the mean of that distribution to represent the image, and then compute the most likely label on the basis of the likelihood of the mean vector belonging to one of the labels. To get good classification performance, the model must learn a latent space representation that correctly separates the labels.

## 2.1 Enhancing the Decoder with label information

In the basic PSE model, the decoder takes as input a sampled latent variable $z$ and produces an image $\hat{x}$. There is however no explicit discriminative supervisory signal given to the decoder during training relating to the target label of the image being generated. Inspired by the approach taken in InfoGAN (Chen et al., 2016), we can extend our model to enable label information to aid the training of the image generator. As the entire model is trained end-to-end, the hope is that this addition will both reduce reconstruction loss, and also produce improved semantic alignment (as evidenced by improved classification accuracy) within the latent space.

In our extended model, PSE*, we add an additional classification network to predict a label, $\hat{l}$, from the reconstructed image. That is, $\hat{l} = \text{Cla}(\hat{x}|\boldsymbol{\theta_l})$, where Cla is the classifier network that takes input $\hat{x}$ with weights $\boldsymbol{\theta_l}$. In a similar manner as for an image, we can define $p_{\theta_l}(\boldsymbol{l}|\boldsymbol{z})$ as the probability of generating a label vector $\boldsymbol{l}$ given a latent vector $\boldsymbol{z}$. Following the same rationale we used for images, we choose $p_{\theta_l}(\boldsymbol{l}|\boldsymbol{z}) = p(\boldsymbol{l}|\hat{l})$ to be the probability of generating the correct label $\boldsymbol{l}$ from the predicted label $\hat{l}$. If the labels are encoded as one-hot vectors then we can take $\log(p(\boldsymbol{l}|\hat{l}))$ to be the cross-entropy, while if we are using word2vec embeddings we could for example use $p(\boldsymbol{l}|\hat{l}) = \mathcal{N}(\boldsymbol{l}|\hat{l}, \sigma^2\,\boldsymbol{I})$.

We can then define a modified objective function

$$L_{\text{PSE}^*}(\boldsymbol{x}, \boldsymbol{l}) = \mathbb{E}_{q_{\phi_x}(\boldsymbol{z}|\boldsymbol{x})}\big[\log\big(p_{\theta_x}(\boldsymbol{x}|\boldsymbol{z})\big) + \log\big(p_{\theta_l}(\boldsymbol{l}|\boldsymbol{z})\big)\big] - D_{\text{KL}}\big(q_{\phi_x}(\boldsymbol{z}|\boldsymbol{x})\,\big\|\,q_{\phi_l}(\boldsymbol{z}|\boldsymbol{l})\big). \quad (3)$$

This extended model provides a further layer of auto-encoding (i.e. we generate an image $\hat{x}$ and a label $\hat{l}$) which, as we see in Section 4, provides a further improvement in performance. The new term $\mathbb{E}_{q_{\phi_x}(\boldsymbol{z}|\boldsymbol{x})}\big[\log\big(p_{\theta_l}(\boldsymbol{l}|\boldsymbol{z})\big)\big]$ can be considered as the mutual information between labels and images, as introduced by Chen et al. (2016).

In these models the labels need to be encoded in vector form. This can be a simple one-hot encoded vector that the model embeds or it could be encoded using a standard pre-trained word embedding such as word2vec. For short text description and a suitable embedding we can average the embedding vectors for each word. As most word embeddings are high dimensional the average of a few words usually corresponds to a vector whose nearest neighbours are the words being averaged. As an example, if we take the phrase "white horse" and calculate the mean of the word2vec embeddings for "white" and "horse" we obtain a vector in the word2vec space whose two nearest neighbour words (measured using cosine similarity) are "white" and "horse".

## 3 Related Work

The area of building visual-semantic encodings that link visual images with labels has a long history in the fields of image retrieval and annotation (e.g. Hare et al., 2006; 2008). Recently, as a result of advances in deep learning, it has become a highly active area of research (e.g. Frome et al., 2013; Akata et al., 2013; Karpathy et al., 2014; Akata et al., 2016; Reed et al., 2016; Zhang et al., 2017). These models have been applied to zero-shot attribute prediction (Frome et al., 2013; Akata et al., 2013), image annotation and text-based image retrieval (Karpathy et al., 2014; Karpathy & Fei-Fei, 2015; Reed et al., 2016). An exciting advance in some of the recent models is that they are fully generative; that is they model a distribution in the latent embedding space that can be sampled and used to generate novel images.

Our proposed PSE and PSE* models fall in the latter category of generative models. We argue that our models are both simpler than existing models, but also more flexible, allowing us to trivially couple to a word2vec embedding rather than one-hot encoded labels. At the same time, our models do not impose independence constraints on the textual labels, and actually represent embedded text or labels by probability distributions in the same latent space as the image distributions. The remainder of this section explores the state-of-the-art in visual-semantic embedding models.

### Visual-semantic embedding.

The common approaches for visual-semantic embedding models learn semantic relationship between images and annotated text descriptions into a vector space by maximising visual and semantic

similarity. The Deep visual-semantic embedding model (DeViSE), in which the core visual model and language model are pre-trained, uses a combination of dot-product similarity and hinge rank loss (Frome et al., 2013). DeViSE has shown the ability to make correct predictions across unseen classes by leveraging semantic knowledge. Karpathy et al. (2014) introduced a structured max-margin objective to calculate the similarity between images and sentences. Reed et al. (2016) propose a structured joint embedding model for end-to-end training to improve the performance on fine-grained visual recognition.

To represent text descriptions as abstract concepts, some works use Gaussian distributions to model words. Ren et al. (2016) use Mahalanobis distance to measure the distance between visual features and text distributions. Wang et al. (2017) propose a VAE-based visual-semantic models to learn label distributions from visual features, in which a marginal regulariser is introduced to separate one label distribution to its nearest neighbour distribution maximally. However, these studies all aim at performing image classification; they are not capable of generating visual contexts after training.

## VAE-BASED CONDITIONAL GENERATIVE MODELS AND JOINT GENERATIVE MODELS.

There are a number of models that are similar in design to ours, and that we can directly compare against. In most cases the labels in these models are taken to be independent and thus these models cannot capture label dependence or similarity, or be easily extended to using free text descriptions. Nevertheless we can compare the performance of our model against these using one-hot encoded labels.

**CVAE.** *Conditional Variational Auto-Encoders* (CVAEs) extend the original ELBO to a conditional bound $\text{elbo}(\boldsymbol{x}|\boldsymbol{l})$ to approximate the conditional log-likelihood, when the encoder and decoder are conditioned with label information (Kingma et al., 2014; Sohn et al., 2015). The standard CVAEs assume that all the prior label embeddings for different classes share the same Gaussian prior. We can view the CVAE latent space as being extended by a one-hot vector encoding the actual label. The model thus directly assumes independence of the labels as they are orthogonal in the latent space. The objective of the standard CVAE has the form

$$L_{\text{CVAE}}(\boldsymbol{x}, \boldsymbol{l}) = \mathbb{E}_{q_\phi(\boldsymbol{z}|\boldsymbol{x}, \boldsymbol{l})}\big[\log\big(p_\theta(\boldsymbol{x}|\boldsymbol{z}, \boldsymbol{l})\big)\big] + D_{\text{KL}}\big(q_\phi(\boldsymbol{z}|\boldsymbol{x}, \boldsymbol{l}) \,\|\, p(\boldsymbol{z})\big). \tag{4}$$

where $p(\boldsymbol{z})$ is a Gaussian prior for all latent distributions. When a latent code $\boldsymbol{z}$ is combined with different labels represented as one-hot vectors, images with various labels can be generated. Therefore, the learned part of the embedding space (excluding the one-hot label dimensions), does not contain any label-specific semantic information. For CVAEs, modality information cannot be exchanged bidirectionally, because learnt latent vectors cannot be used to do image annotation.

**JMVAE and TrELBO.** The *Joint Multimodal Variational Auto-Encoder* (JMVAE, Suzuki et al. (2016)) and the Triple ELBO model (TrELBO, Vedantam et al. (2018)) learn the joint representation from multiple inputs with complicated training objectives. Both models just concatenate images and labels represented by one-hot vectors as the input of image and label encoder $q(\boldsymbol{z}|\boldsymbol{x}, \boldsymbol{l})$. However, these models still keep a Gaussian prior. Therefore, joint embedding distributions are forced to be closed to the prior, which causes more overlap in the embedding space which can be disadvantageous as dissimilar concepts might be forced closer together. The objective of JMVAE has the form

$$\begin{aligned} L_{\text{JMVAE}}(\boldsymbol{x}, \boldsymbol{l}) = &\mathbb{E}_{q_\phi(\boldsymbol{z}|\boldsymbol{x}, \boldsymbol{l})}\big[\log\big(p_{\theta_{\boldsymbol{x}}}(\boldsymbol{x}|\boldsymbol{z})\big)\big] + \mathbb{E}_{q_\phi(\boldsymbol{z}|\boldsymbol{x}, \boldsymbol{l})}\big[\log\big(p_{\theta_{\boldsymbol{l}}}(\boldsymbol{l}|\boldsymbol{z})\big)\big] \\ &- D_{\text{KL}}\big(q_\phi(\boldsymbol{z}|\boldsymbol{x}, \boldsymbol{l}) \,\|\, p(\boldsymbol{z})\big) \\ &- \alpha\left(D_{\text{KL}}\big(q_\phi(\boldsymbol{z}|\boldsymbol{x}, \boldsymbol{l}) \,\|\, q_{\phi_{\boldsymbol{x}}}(\boldsymbol{z}|\boldsymbol{x})\big) + D_{\text{KL}}\big(q_\phi(\boldsymbol{z}|\boldsymbol{x}, \boldsymbol{l}) \,\|\, q_{\phi_{\boldsymbol{l}}}(\boldsymbol{z}|\boldsymbol{l})\big)\right). \end{aligned} \tag{5}$$

This objective contains two reconstruction terms and three regularisation terms. $D_{\text{KL}}(q_\phi(\boldsymbol{z}|\boldsymbol{x}, \boldsymbol{l})\|q_{\phi_{\boldsymbol{l}}}(\boldsymbol{z}|\boldsymbol{l}))$ makes the JMVAE learn a joint visual-semantic embedding space. However, this model does not embed labels to a latent space entirely based their visual contexts. There is still a potential intersection between different labels which is not caused by visual similarity.

The TrELBO introduces two unpaired inference networks to fit a joint distribution to learn distributions of visual concepts. The objective is

$$
\begin{aligned}
L_{\text{TrELBO}}(\boldsymbol{x}, \boldsymbol{l}) = {} & \mathbb{E}_{q_\phi(\boldsymbol{z}|\boldsymbol{x},\boldsymbol{l})}\left[\log\left(p_{\theta_{\boldsymbol{x}}}(\boldsymbol{x}|\boldsymbol{z})\right)\right] + \lambda_1 \mathbb{E}_{q_\phi(\boldsymbol{z}|\boldsymbol{x},\boldsymbol{l})}\left[\log\left(p_{\theta_l}(\boldsymbol{l}|\boldsymbol{z})\right)\right] \\
& + \mathbb{E}_{q_{\phi_{\boldsymbol{x}}}(\boldsymbol{z}|\boldsymbol{x})}\left[\log\left(p_{\theta_{\boldsymbol{x}}}(\boldsymbol{x}|\boldsymbol{z})\right)\right] + \lambda_2 \mathbb{E}_{q_{\phi_l}(\boldsymbol{z}|\boldsymbol{l})}\left[\log\left(p_{\theta_l}(\boldsymbol{l}|\boldsymbol{z})\right)\right] \\
& - D_{\text{KL}}\left(q_\phi(\boldsymbol{z}|\boldsymbol{x},\boldsymbol{l}) \,\big\|\, p(\boldsymbol{z})\right) \\
& - D_{\text{KL}}\left(q_{\phi_{\boldsymbol{x}}}(\boldsymbol{z}|\boldsymbol{x}) \,\big\|\, p(\boldsymbol{z})\right) - D_{\text{KL}}\left(q_{\phi_l}(\boldsymbol{z}|\boldsymbol{l}) \,\big\|\, p(\boldsymbol{z})\right).
\end{aligned}
\tag{6}
$$

This objective is more complicated, but it cannot learn a compositional latent embedding space, because it lacks a regulariser to learn the relationship between label distributions and image distributions.

**SCAN.** Symbol-Concept Association Network (SCAN, Higgins et al. (2018)) is capable of learning representation of concepts from visual content. SCAN extends $\beta$-VAE (Higgins et al., 2017), which can learn disentangled latent representations. Concept distributions are learnt from these disentangled representations. Therefore, SCAN does not learn a visual-semantic embedding space directly. In addition, SCAN does not generate high-quality image samples from symbolic concepts, because of the trade off between reconstruction quality and disentanglement in $\beta$-VAE (the controlled capacity term is not applied in the training process of SCAN (Higgins et al., 2018; Burgess et al., 2018)). SCAN is not an end-to-end model; it uses a $\beta$-VAE-based model (Higgins et al., 2017) to learn disentangled latent representations, and then it uses the KL divergence $D_{\text{KL}}(q_{\phi_{\boldsymbol{x}}}(\boldsymbol{z}|\boldsymbol{x})\|q_{\phi_l}(\boldsymbol{z}|\boldsymbol{l}))$ to learn average label distributions to fit disentangled image distributions with the same labels. However, learning disentangled representations in this way punishes reconstruction quality.

## 4 Experiments with the PSE and PSE* models

In this section, we firstly evaluate our label embedding model on two tasks: label-to-image generation and image annotation. Models we compare are the standard CVAE, JMVAE, TrELBO and SCAN. We train these models on MNIST (LeCun & Cortes, 2010) and Fashion-MNIST (Xiao et al., 2017), which both consist of 50 000 grayscale $28 \times 28$ images for training and 10 000 examples as the testing set. Each image is associated with a label from 10 classes of digits (0-9) or clothing items ('T-shirt/top', 'Trouser', 'Pullover', 'Dress', 'Coat', 'Sandal', 'Shirt', 'Sneaker', 'Bag', 'Ankle boot'). Labels are encoded as one-hot vectors.

We use MLPs for the encoders and decoders of all these models, trained using the Adam optimiser with a learning rate of 0.001. The models are trained with 2000 steps with a minibatch size of 100. Full details of the model architectures and hyperparameter settings are described in Appendix A.

On the MNIST dataset, we test all the training objective without any training tricks. Specifically we do not use warm-up (Sønderby et al., 2016), inference network freeze (Vedantam et al., 2018) and denoising auto-encoder L2 loss term replacement (Higgins et al., 2018). These are likely to improve performance in all the models we test. We evaluate the performance of these models on 2D and 10D embedding space respectively. Note that for the CVAE model we consider the dimensionality of the learned part of the latent space, without the extension of the one-hot dimensions. On the Fashion-MNIST dataset, we show the ability of our model to capture visual context changes.

### 4.1 Experiments on MNIST

**Embedding space visualisations.** For visualising the embedding space we use a 2-dimensional latent space. Although this gives relatively poor reconstructions it allows us to observe how different objectives influence the learnt embedding space (see Figure 2). It is apparent that the standard CVAE does not learn any discriminant label distributions within the learnt part of the latent vector. The embedding space is consistent with its assumption for label distributions: $p(\boldsymbol{z}|\boldsymbol{l}) = p(\boldsymbol{z})$.

The embedding spaces of the JMVAE and the TrELBO are similar. Compared with the VAE, latent codes for images with the same labels are clustered. However, the Gaussian prior forces all the clusters to be squeezed to the prior's mean. The latent space of SCAN is encoded by a pre-trained $\beta$-VAE.

In the embedding space of PSE and PSE*, we find that distributions with different semantics are no longer located around a prior's centre as is the case in the VAE, JMVAE and TrELBO models. This gives us a more separable semantic latent space. With the relaxed prior assumption, label embeddings only depend on relative visual features, and each individual label distributions can keep the shape of a Gaussian. These characteristics make it possible for semantic knowledge to be induced in the learnt latent spaces of PSE and PSE*. Compared with PSE, PSE* has a latent space in which there are clearer separation of different labels. In Figure 2f, we can see that the label '9' is embedded in the middle of the label '4' and the label '7', which demonstrates that PSE* learns an embedding space capturing visual similarities. Appendix B has a larger visualisation of the spaces generated by both variants of our model for both MNIST and Fashion-MNIST.

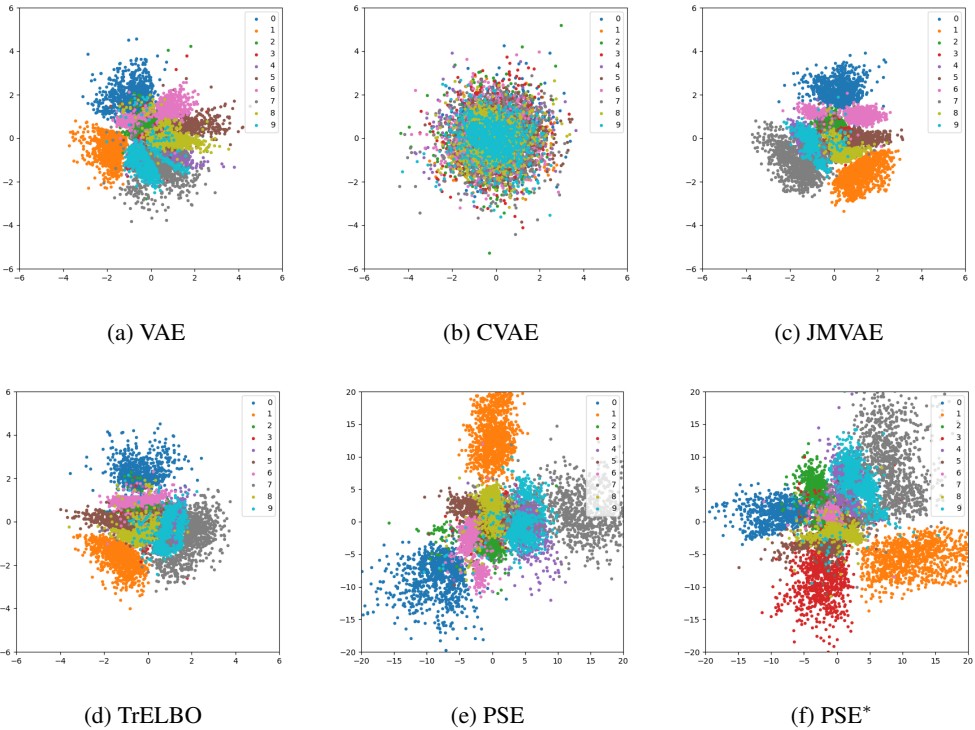

Figure 2: 2D embedding space visualisations of MNIST characters. Best viewed in colour. Points are the latent image embedding codes, and different colours represent different labels. We can see that the *learned* latent dimensions of CVAE does not obtain a semantic embedding space. The JMVAE, the TrELBO model and the SCAN (in SCAN, images are encoded with a pre-trained $\beta$-VAE.) force learnt latent distributions to match with the Gaussian prior; therefore there is serious overlap in the embedding space. Our PSE and PSE* label embedding models provide better separation of classes whilst capturing visual similarity, with PSE* showing the greater separation of classes. See Appendix B for a larger visualisation of the PSE and PSE* latent spaces.

**Quantitative evaluation.** In this part, we evaluate our model, the CVAE, the JMVAE and the TrELBO model by measuring bidirectional generation accuracy and embedding performance. Compared with the evaluation methods in other recent work (Vedantam et al., 2018), we do not choose the model parameters which maximise an overall performance metric on a validation set. Instead we aim to set these parameters equivalently across all models for fair comparison. We also show the influence of the additional loss term in our extended PSE* model.

To measure the accuracy of generated images, we pre-trained an MLP classifier on the MNIST training data. This classifier reached a classification accuracy of 98.09% on training data and an accuracy of 96.86% on the validation set. For each model being explored, 10 000 samples per label distribution are drawn, and used to generate 100 000 testing images. The pre-trained classifier is used

to test whether each of the 100 000 generated test images could be correctly classified as belonging to the label it was generated from. Higher classification accuracy across these generated images (which we term 'Generation Accuracy') should indicate better quality reconstructions (although high values can also be indicative of the models producing less diverse images). This is repeated 10 times to compute standard deviations. We also assess the quality of generation directly by measuring the reconstruction error on the MNIST test data.

As described in Section 2, for the image annotation task we calculate the probability that an encoded image (represented by the mean vector of its probability distribution) belongs to a specific label distribution. The annotation accuracy can be considered as a metric to measure embedding performance. A higher accuracy illustrates a more discriminable embedding space.

Table 1: Evaluation of different models on the MNIST test dataset

| Model | Reconstruction Error | Annotation Accuracy (%) | Generation Accuracy (%) |
|---|---|---|---|
| CVAE | **79.8** | N/A | **97.6** $\pm$ 0.1 |
| JMVAE | 87.7 | 95.4 | 92.2 $\pm$ 0.3 |
| TrELBO | 83.1 | 90.3 | 61.1 $\pm$ 0.5 |
| PSE (10D) | 82.8 | 96.0 | 93.9 $\pm$ 0.2 |
| PSE$^*$ (10D) | 82.3 | 96.3 | 94.9 $\pm$ 0.2 |
| PSE$^*$ (20D) | **75.0** | **97.1** | 95.6 $\pm$ 0.2 |

From Table 1, we can see that the standard CVAE has achieved the lowest reconstruction error and the highest generation accuracy compared with other models with 10-dimensional latent space[1]. However, CVAE can not perform image annotation as it does not have the ability to learn label distributions in the same space as the visual distributions. We see that the TrELBO has a better reconstruction performance compared with the JMVAE, while it leads to less correct generated samples. When $\lambda_1 = 1$ in the TrELBO, the conditional generation accuracy is the lowest ($61.0 \pm 0.5$), this trend is similar to that found by Vedantam et al. (2018). Figure 3 shows samples generated by (a) the TrELBO and (b) the PSE$^*$ model.

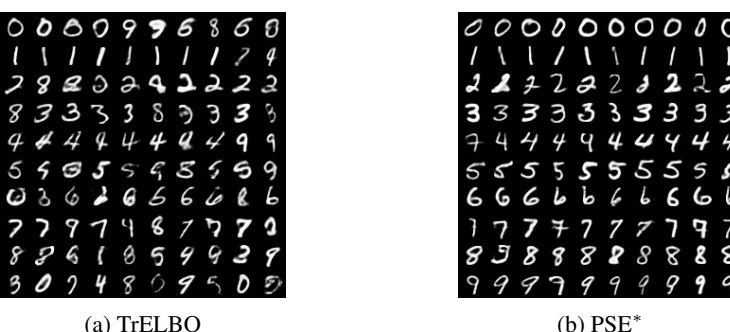

(a) TrELBO                    (b) PSE$^*$

Figure 3: Generated samples of the TrELBO model and our model. We see that the TrELBO can not create accurate images without a higher hyperparameter $\lambda_1$.

We do three experiments using our models: Firstly, we explore the use of the PSE model. Then we evaluate and compare against the objective of the extended PSE$^*$ model. Finally, we extend the dimension of latent embedding space to 20, to make it more comparable with the CVAE (considering that in CVAE the conditional label code is part of the latent representation). We see that our model has the best performance in both conditional image synthesis and image annotation, compared with the JMVAE and the TrELBO. Using the extended PSE$^*$ model to incorporate label information is shown to help obtain a more meaningful embedding space (the space is more discriminable). We also see that a higher dimensional latent space can both improve the performance on label-to-image and image-to-label tasks.

---

[1]technically the CVAE has a 10+10 dimensional space if we consider the concatenation of the one-hot label vector required to drive the generator and should thus more correctly be compared to the 20D models.

## 4.2 EXPERIMENTS ON FASHION-MNIST

We next test our model on Fashion-MNIST by exploring visual changes on the learnt embedding space, as well as performing a quantative comparative evaluation.

Results of the quantative evaluation are shown in Table 2. We use the same methodology to evaluate image reconstruction and annotation performance as we do on MNIST. For the generation accuracy, we used a pre-trained a ResNet-18 (He et al., 2016) with a classification accuracy of 94.5% on training data and an accuracy of 92.4% on the validation set. We find that our model also has the best performance in both conditional image synthesis and image annotation, compared with JMVAE and the TrELBO. With supervised information, the extended PSE* also shows a better label-to-image and image-to-label performance. Interestingly all variants of our models outperform CVAE, even when handicapped with a lower effective dimensionality.

Table 2: Evaluation of different models on the Fashion-MNIST test dataset.

| Model | Reconstruction Error | Annotation Accuracy (%) | Generation Accuracy (%) |
|---|---|---|---|
| CVAE | 219.7 | N/A | $\mathbf{81.3 \pm 0.3}$ |
| JMVAE | 225.4 | 80.3 | $71.2 \pm 0.5$ |
| TrELBO | 220.3 | 73.0 | $47.5 \pm 0.4$ |
| PSE (10D) | 219.1 | 81.2 | $75.9 \pm 0.3$ |
| PSE* (10D) | 219.6 | 81.3 | $76.1 \pm 0.4$ |
| PSE* (20D) | $\mathbf{219.0}$ | $\mathbf{81.6}$ | $76.2 \pm 0.4$ |

In the final section, we observe that all the label distributions gather together to the centre, because most VAE-based joint learning multimodal models assume a single Gaussian prior. This assumption causes more overlap in the embedding space, and it also breaks the alignment of label distributions, which is based on visual features. With this assumption, our model can learn gradual visual context changes from one label to its visual-semantic neighbour. For example, boots and sneakers are visual-semantic neighbours and the difference between them are the heights around the ankles. A visual-semantic embedding space should be able to capture these changes. To illustrate this property, we sample latent codes along the line through the means of two neighbouring label distributions. Drawn samples are used to generate a series of images between two labels. Figure 4b shows the changes from sneaker to boot and the changes from T-shirt to shirt. The results demonstrate that label distributions learnt by our model is only aligned by visual contexts.

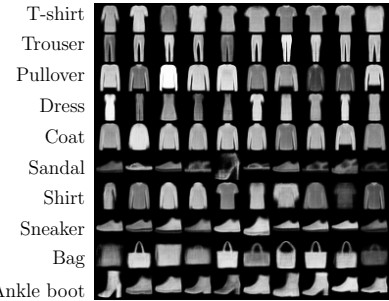

(a) Conditional image generation on Fashion-MNIST

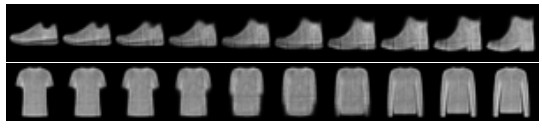

(b) Sneaker to Ankle Boots & T-shirt to Shirt

Figure 4: Our model can learn the gradual visual changes. By exploring the embedding space, a series of images can be generated to illustrate how one class of images change to their visual-semantic neighbours.

## 5 EXPLOITING THE SEMANTICS OF A PRE-TRAINED WORD EMBEDDING SPACE

In the previous section, all the experiments used a one-hot encoding of the image labels as input to the label encoder, $\phi_t$. In this case, the label encoder would learn the probablistic embeddings of

these vectors by purely conditioning on what is learnt by the image encoder; in other words, in the probabilistic embedding space learned by our models, label proximity would be driven by visual similarity. An obvious extension is to explore if prior knowledge of word similarity can help better structure the latent representation, and offer additional affordances (for example by allowing our model to generate images for labels that are related to, but outside of the vocabulary of image labels used in training the model). In the following, we explore this extension, and show that a pre-trained word embedding can provide prior semantic information to our proposed visual-semantic embedding model, allowing us to imagine unseen images. In this sense, we illustrate our model's ability for zero-shot learning.

For the experiments presented here, we use pre-trained word2vec vectors[2] instead of one-hot vectors as input label representations to explore the influence of prior text knowledge for visual-semantic embedding. We note that the particular choice of pre-trained word embedding is arbitrary, and that more recent models such as FastText (Bojanowski et al., 2016) and ELMo (Peters et al., 2018) could equally be used, and might have better performance. For the purposes of the experiments presented here, the word2vec model suffices to demonstrate the idea however.

Compared with the one-hot vectors, word2vec vectors are continuous. Label representations have different initial semantic distances, with similar-meaning (in terms of what can be captured from the common contexts in which they occur) words being closer together. As with one-hot vectors, we can capture visual changes and perform inference by exploiting the word embedding space directly when we input word2vec vectors into the model. Figure 5 shows the generated images from two word2vec label representations and inferred images from averaged vectors in the pre-trained embedding space. As shown in Figure 4b, we can see that the inferred clothing items contain visual features of their two neighbours.

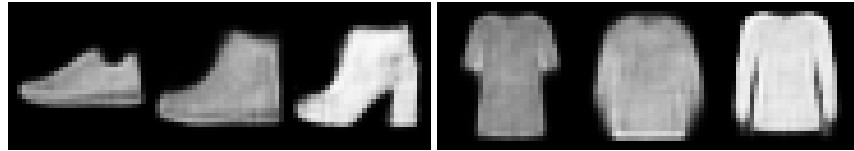

Figure 5: Images generated from a PSE* model trained with word2vec encoded labels and label compositions. Images represent: 'Sneaker', 0.5('Sneaker'+'Ankle Boots'), 'Ankle Boots' & 'Tshirt', 0.5('Tshirt'+'Shirt'), 'Shirt'

With pre-trained word2vec, prior knowledge learnt from the text domain can be exploited to imagine images with unseen labels — we can generate images for terms that were not used in the training set. Figure 6 shows six examples generated from words semantically close to, but different from, the training labels.

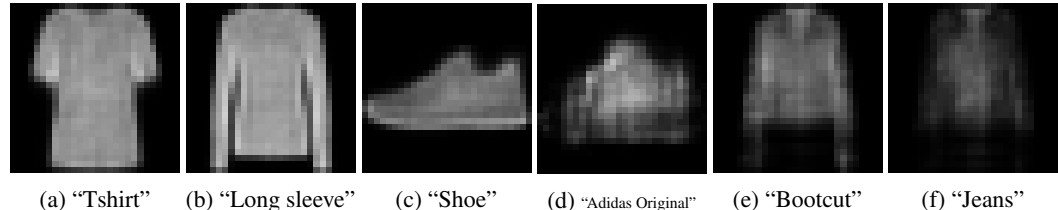

(a) "Tshirt"  (b) "Long sleeve"  (c) "Shoe"  (d) "Adidas Original"  (e) "Bootcut"  (f) "Jeans"

Figure 6: Generated samples for unseen labels

We can see that our models can leverage the similarity in the text domain and generate relevant visual features. We find that prior knowledge from a pre-trained word2vec contains some noise for visual-semantic information. For example, in the image generated in Figure 6e for the term "Bootcut" could be considered to be incorrect, as we would most likely expect an image that looks like a pair of trousers. However, in the pre-trained word2vec model, "Bootcut" is most similar to "Pullover". This demonstrates that prior knowledge is exploited by visual-semantic embedding. This is also

---

[2]Using the standard 300-dimensional pre-trained embedding available from `https://code.google.com/archive/p/word2vec/`.

Table 3: Performance using one-hot vectors and word2vec vectors on the testing dataset of Fashion-MNIST with the PSE and the PSE* models

| Label Encoding Schemes | Reconstruction Error | Annotation Accuracy (%) |
|:---:|:---:|:---:|
| one-hot(PSE) | 219.1 | $81.2 \pm 0.1$ |
| word2vec(PSE) | **218.7** | $81.4 \pm 0.2$ |
| one-hot(PSE*) | 219.6 | $81.3 \pm 0.2$ |
| word2vec(PSE*) | 218.8 | **81.9** $\pm 0.3$ |

further demonstrated in Figure 6f ("Jeans"), whose word2vec nearest neighbours include a variety of clothing items and the general term "Clothing".

**The effect of using word2vec inputs on objective performance metrics.** Table 3 shows the performance differences for Fashion-MNIST reconstruction error and classification accuracy when using the PSE and the PSE* models with one-hot vector and word2vec vector inputs. Interestingly we see that using pre-trained word embeddings can help achieve small, but statistically significant ($p = 0.006$ for the annotation accuracy on the PSE* method), improvements in reconstruction and annotation results. This demonstrates that introducing prior textual knowledge into our model can provide useful semantic information for visual-semantic embedding.

## 6 CONCLUSIONS AND FUTURE DIRECTIONS

We propose an extension of VAEs that allows us to couple different modalities through a shared latent space. In the paper, we have coupled images with labels. Because each label has a large number of different images associated with them, we can encode each label by a broad probability distribution. We replace the priors in VAEs by these label distributions. This allows the latent space to self-organise creating a semantically meaningful representation of the data.

Although this is a relatively simple modification of VAEs, we believe it to be huge powerful. We have demonstrated some of the ability this latent space representation provides, but we have only scratched the surface. At present we have used very simple set of images with very limited labels. Clearly an important next step is to use a richer set of images with richer text descriptions. In particular, we have not yet studied the use of attribute descriptions that potentially allows much more powerful semantic space representation. Using pre-trained word embeddings such as word2vec solves the problem of synonymy (different words having the same meaning), but, because each word is mapped to a point, the representation cannot cope with polysemy (the same word having different meanings). In our model we perform a second embedding of the text to the combined image-text latent space, but this time the embedding is probabilistic. This provides a potential solution to polysemy in that words such as "bank" with multiple meanings can be represented as a broader probability distribution, or a probability distribution with multiple modes, than very specific words. It would require a richer training set to test whether the PSE and PSE* models capture these linguistic subtleties.

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

## A    MODEL DETAILS

**PSE and PSE**$^*$**.** The image encoder, $q_{\phi_x}(z|x)$, is an MLP with 512 and 64 hidden units. The label encoder, $q_{\phi_l}(z|l)$, consists of one dense layer. The image decoder, $D_l(\hat{x}|\theta_l)$, is an MLP with 64, 512 hidden units. The decoder used is Bernoulli following Suzuki et al. (2016) and Higgins et al. (2018). The classifier, $\text{Cla}(\hat{x}|\theta_l)$, is parameterized as an MLP with 512, 64 hidden units. ReLU activation is used throughout. We use the Adam optimizer with a learning rate of 0.001. The model is trained with 2000 steps with a minibatch size of 100.

**CVAE.** In a standard CVAE, we concatenate labels with flattened images and pass them to an encoder parameterized as an MLP with 512, 64 hidden units. And then labels concatenated with latent vectors are passed to an MLP decoder with 64, 512 hidden units. ReLU is also used in the standard CVAE. The training epochs and minibatch size are the same as PSE.

**JMVAE, TrELBO.** JMVAE and TrELBO have the image encoders and label encoders with a similar architecture with the PSE. For their joint encoder, labels are concatenated with flattened images which are passed to an MLP with 512, 64 hidden layers. ReLU is used as the activation function. The training epochs and minibatch size are the same as PSE.

**SCAN.** SCAN architecture is based on a $\beta$-VAE, which has an encoder parameterized as an MLP with 512, 64 hidden units and an MLP decoder with 64, 512 hidden units. In our case, $\beta$ is 10. The training epochs and minibatch size are the same as PSE.

## B    THE EMBEDDING SPACE OF PSE ON MNIST

The following illustrations (Figures 7-10) show 2D semantic embedding spaces for our PSE and PSE$^*$ models on the MNIST and Fashion-MNIST datasets.

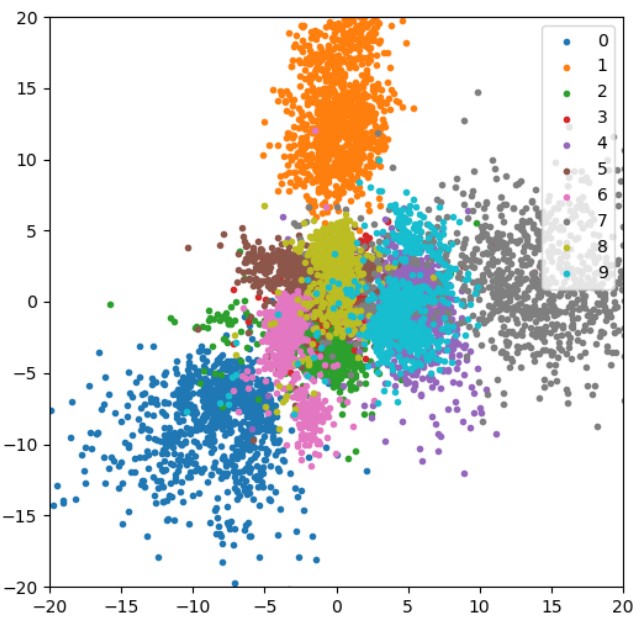

Figure 7: 2D latent space of MNIST with the PSE model.

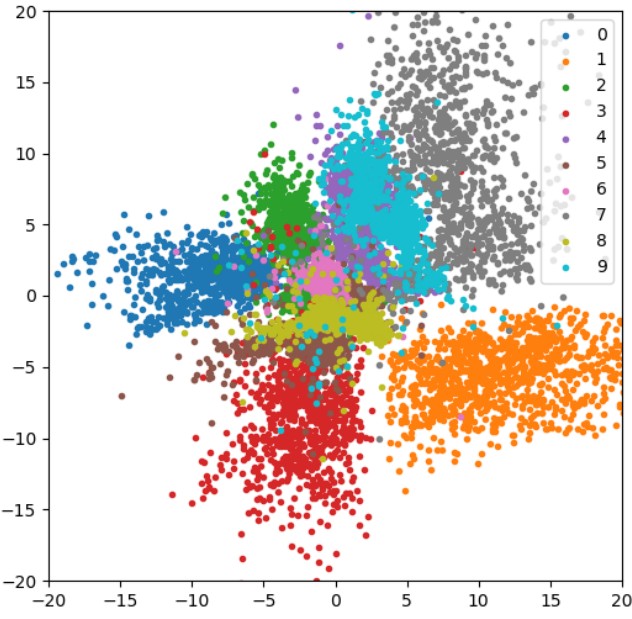

Figure 8: 2D latent space of MNIST with the PSE* model.

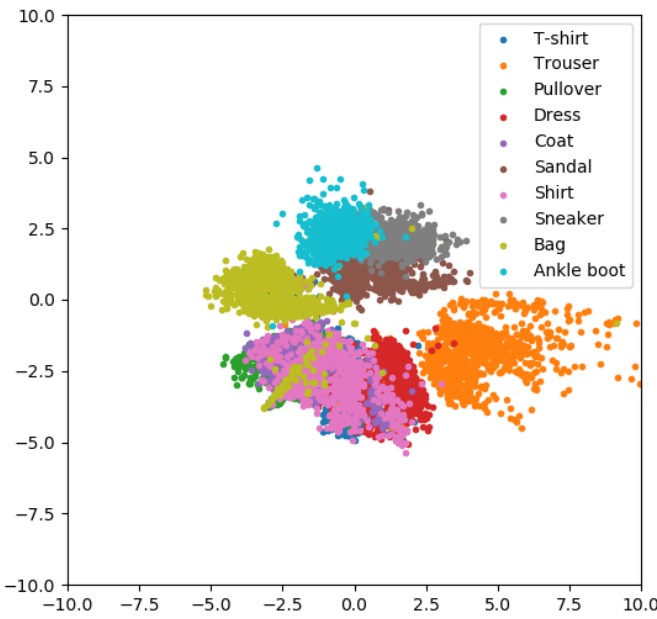

Figure 9: 2D latent space of Fashion-MNIST with the PSE model.

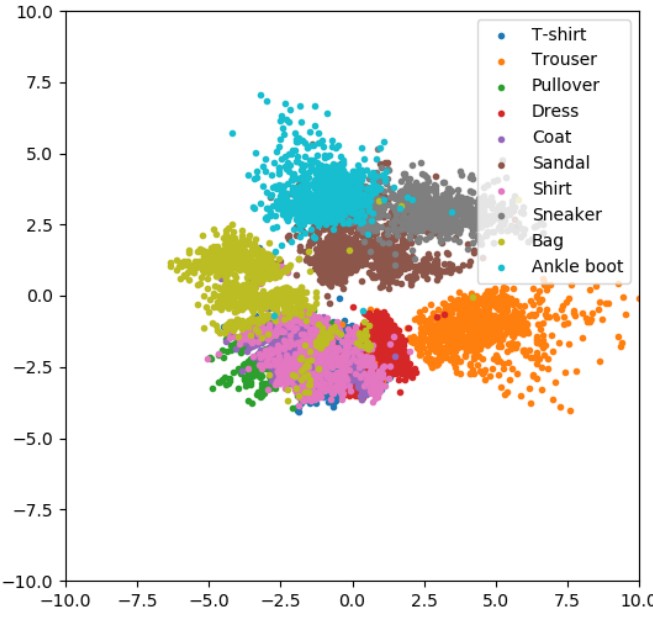

Figure 10: 2D latent space of Fashion-MNIST with the PSE* model.

