# OpenReview forum: "Probabilistic Semantic Embedding"
_ICLR.cc/2019/Conference_

### Official Review · AnonReviewer3 · 2018-11-02
**A simple extension of VAE**

**Rating:** 4
**Confidence:** 4

**Review:**

This paper proposes to replace the traditional KL term of VAE with the KL between two conditional distributions, which is to equip the model with the ability to address multimodal data. Moreover, the paper also extend the model to add an additional network to predict the label from the reconstructed image to enhance the decoder with supervised information.

In general, the model is contrived and the  novelty of the paper is incremental. Is Eq. 2 the ELBO of the new model? If so, the authors should provide the derivation of the ELBO. If not, can you prove the objective is the correct one to be optimized?
Moreover, the model is just an trivial extension of VAE and all key techniques are borrowed from existing work (Chen et al. 2016).

The experiments are only conducted on MNIST and Fashion-MNIST, which is not sufficient. CIFAT10 should be at least added, and other more challenging benchmarks should also be considered to make the model more solid.

---

> ### Author Response · Authors · 2018-11-20
> **Response to R3[1/2]**
>
> We would like to thank the reviewer for spending time to read our paper. We would like to use this opportunity to clarify the misunderstandings that the reviewer had from their initial reading.
>
> "In general, the model is contrived and the  novelty of the paper is incremental. Is Eq. 2 the ELBO of the new model? If so, the authors should provide the derivation of the ELBO. If not, can you prove the objective is the correct one to be optimized?
> Moreover, the model is just an trivial extension of VAE and all key techniques are borrowed from existing work (Chen et al. 2016)."
>
> We contest the statement that the model is contrived.  This seems to be based on the assumption that we are attempting to fulfil the same objective as the original VAE paper of maximising the likelihood of generating the test set, rather than finding a probabilistic embedding of both images and classes (words or labels) such that the geometry of the embedding space is semantically meaningful (as the title of the paper suggests).  The original VAE paper leaves many open questions, such as what is the role of the prior? (does it really encode our prior belief?).  A more satisfactory interpretation of VAEs, at least for what we are trying to do, is in terms of minimum description length.  That is, it is a machine for efficiently communicating the input x by sending a message z and the error \epsilon = x - \hat{x}.  This is more efficient than directly sending x as z is low dimensional and can be communicated to low accuracy (as determined by q(z|x)) and \hat{x} is much more sharply concentrated than x.  The objective function -log(p(\epslion)) + E_z[ KL(q||p) ] is proportional to the expected message length: the first term is the length of message to communication the error (up to some accuracy -log(\Delta) that is the same for any model) and the KL-divergence or relative entropy measures the length (in nats) of the message required to communicate z with uncertainty q assuming the messages have probability p(z).  Minimising the VAE objective function can be viewed as directly minimising the message length.
>
> Within this formulation you can ask what should the underlying distribution p(z) be.  In particular if you have classes with different variation it would seem natural to learn the "priors" based on the data. (Learning priors in a Bayesian framework would require introducing hyper-parameters and using the evidence to determine them---often computationally expensive.  But in a minimum description framework we would can learn these "priors" more naturally by minimising a combination of log errors and KL divergences as we have done).  Minimising the PSE objective function can be viewed as simultaneously learning both the embeddings and priors for labels to minimise the expected message length. Thus far from being contrived we would argue that our model is very natural (much more so than models like CVAE).
>
> We would also strongly contest that the paper is incremental.  We have learned a very non-trivial embedding based on a combination of visual similarity and labels (or the semantics inherited from the word2vec embedding).  This allows us to perform tasks such as generating meaningful regions in our latent space from words (or short descriptions) that we have not seen during training.  This is not an incremental improvement on previous work but fundamentally novel!
>
> We concede that PSE* is based on an idea borrowed from Chen et al. (as we fully acknowledged in the paper) but it is a small modification to the proposed PSE model which is the core of our paper.  We included it as, to the best of our knowledge, it is the first time that the idea has been used with VAEs and it provided a small but significant improvement in performance.

---

> ### Author Response · Authors · 2018-11-20
> **Response to R3[2/2]**
>
> "The experiments are only conducted on MNIST and Fashion-MNIST, which is not sufficient. CIFAT10 should be at least added, and other more challenging benchmarks should also be considered to make the model more solid."
>
> We accept that it is desirable to include additional data sets (something we are currently working on, but that takes time).  Clearly if we were, for example, proposing a new classifier then results on MNIST would clearly be inadequate.  However, for exploring the semantics of latent spaces, MNIST and Fashion-MNIST provide an adequate proof of principle in our judgement.  We cannot see why the latent space embedding would be very different if we used a data set that is visually more complex.  It is unclear to us that the CIFAR-10 data set is the best next step as many of the classes in CIFAR10 seem unrelated (ship and horse) so the learnt semantic embedding is likely to be uninteresting.  CIFAR-100 seems more appropriate, but the relative sparseness of the data makes it big step beyond the MNIST data sets we used. Moving to such a dataset also would require significant changes to the architecture of the encoder and decoder (they would both likely have to be convolutional to achieve good performance), and it is not obvious that this actually adds much to the main point of our paper (which is to demonstrate that our model learns meaningful latent spaces).
>
> To conclude we would once again like to thank the reviewer for their time. In the revised version of the paper, we have addressed the points made by the other reviewers, and hopefully the exposition of the novelty should be even more clear in the revision. We would be happy to respond to any further questions that might arise.

---

### Official Review · AnonReviewer1 · 2018-11-02
**This paper presents a VAE model that jointly models images and their labels.**

**Rating:** 4
**Confidence:** 4

**Review:**



This paper presents a VAE model that jointly models images and their labels. Specifically, the following the VAE framework, the proposed model encodes an image into a latent variable, whose prior is conditioned on the labels of the image, and the latent variable is used to reconstruct the image. The paper also presents a variant which also reconstructs the labels


My comments are as follows:

1. About the significance and originality, although to my knowledge, there seems to be no the exact match in the existing approaches of the idea of incorporating labels into the prior of the latent variable of VAE, the idea seems a little bit trivial and less of technical depth. Moreover, in terms of performance, it seems that the proposed model is not significantly better than the previous models. Therefore, my major concern of this paper goes to the significance.

2. In terms of experiments, I am not convinced that using 20D of PSE VS 10D of CVAE is a fair comparison, although the CVAE will use another 10 dimensions to encode 10 labels with one-hot form. Moreover, using the same settings for all the models in comparison may not be perfect because different models may have different best settings. It would be better if the validation set is used to tune the settings a little bit.

3. Does the proposed model use the word embeddings of the labels? It could be better to report both results of word embeddings and one-hot encoding of the labels, to see if word embeddings help.

---

> ### Author Response · Authors · 2018-11-20
> **Response to R1**
>
> We would like to thank the reviewer for their review. We have addressed each of the three points below, and where appropriate, we also highlight where changes have been made in the revised paper to address these:
>
> 1. About the significance and originality, although to my knowledge, there seems to be no the exact match in the existing approaches of the idea of incorporating labels into the prior of the latent variable of VAE, the idea seems a little bit trivial and less of technical depth. Moreover, in terms of performance, it seems that the proposed model is not significantly better than the previous models. Therefore, my major concern of this paper goes to the significance.
>
> The originality of the model is precisely that to the best of our knowledge no existing model is able to to learn a non-trivial joint probabilistic embedding based on a combination of visual similarity and labels (or the semantics inherited from the word2vec embedding). We demonstrate that our model can both generate images (in a fully-generative sense, by just sampling a latent variable), can generate images conditioned on a label (or even related, but out-of-vocabulary terms when using a word embedding), and can generate labels conditioned on images. The significance is that we can perform all these tasks within a single, well-motivated, model. We further show that the spatial topology of our learned probabilistic space appears to be much more natural than existing models.
>
> 2. In terms of experiments, I am not convinced that using 20D of PSE VS 10D of CVAE is a fair comparison, although the CVAE will use another 10 dimensions to encode 10 labels with one-hot form. Moreover, using the same settings for all the models in comparison may not be perfect because different models may have different best settings. It would be better if the validation set is used to tune the settings a little bit.
>
> No evaluation of these types of models can be perfect as there are just too many hyper-parameters to consider. Our experimental setup tries to keep things as close as possible by using similar choices for the construction of the encoder and decoder networks, and the same values for the latent space dimensionality. All of the models we compare will perform better with larger latent space dimensions or more layers in the encoders and decoders, so it is important that we try to level the playing-field as much as possible to quantify the performance of the models.
>
> With respect to the comparison with CVAE, we would be happy to take on-board any suggestions given by the reviewers as to how to make this fairer. However, at the same time, one has to recognise that by construction, the dimensionality of the input to the decoder of a CVAE is precisely the number of distinct labels plus the number of latent dimensions of the encoder. The CVAE model imposes constraints on the distribution of some of the elements (those belonging to the label) of that input vector, whereas our model does not.
>
> 3. Does the proposed model use the word embeddings of the labels? It could be better to report both results of word embeddings and one-hot encoding of the labels, to see if word embeddings help.
>
> Yes; this is precisely what is described in section 4.3 of the initial review version of the paper (the experiments prior to that use one-hot vectors). Word embeddings give a very slight improvement in objective measures of model performance, but more importantly they endow the model with the ability to generate images terms and phrases for things it has not been trained on (see figure 6 in the original version - none of the query labels for which the images have been generated exist in the data used to train the model). Although not discussed in the paper for space reasons, our model with word embeddings is equally capable of generating potentially relevant words from the word-embedding space given an image. Objective results comparing one-hot against word embeddings is shown in table 2 of the original version of the paper.
>
> In the revised version of the paper we have slightly restructured the text to make it much clearer when we use one-hot encodings and when we use word embeddings. We've also expanded the experimental results using word embeddings to cover both the PSE and PSE* model variants.
>
> Once again, we would like to thank the reviewer for their time. Any further questions, comments or suggestions would of course be gratefully received.

---

### Official Review · AnonReviewer2 · 2018-11-05
**Creating semantic embeddings using textual representations**

**Rating:** 7
**Confidence:** 3

**Review:**

Summary
=========
The authors present an extension to the VAE model by exploring the possibility of using the label space to create a new embedding space, which they call Probabilistic Semantic Embedding (PSE).
They present two different extension of PSE, PSE and PSE*.
The idea of additionally supporting the latent embedding, created by a VAE, by using available textual descriptors seems promising.
The proposed model was evaluated on two tasks, label-to-image generation and image annotation.
Although the work is interesting, there are a few questions that I am not clear about and have several comments.

Questions
=========
1. How was the word2vec model trained? Did you use an existing pretrained model (e.g. available as download) or did you train the embedding model yourself? If so, on what data?
2. The major novelty of this approach is the use of annotations supporting images and textual (pretrained) embedding spaces, but no related work regarding Wes was neither introduced in the Related Work section nor was it clearly explained in the text.
3. Why did the authors focus on the w2v model instead of more promising approaches as fastText or ELMo?
4. How does your model deal with OOV word(s) as input? For example, when used as Image Generator.
5. Table 1 shows results achieved on MNIST but not Fashion-MNIST; was the evaluation performed on MNIST only?
6. Table 2 presents the impact of the use of pretrained embeddings (word2vec) instead of one-hot vectors for labels. Which one do the models presented in Table 1 use?
7. Could you explain the small difference between using one-hot vs pretrained label encodings, presented in Table 2?
8. Also, can you explain how the numbers in table 2 were achieved (e.g. sum over all, average of all, etc.). When comparing the values presented here to the values of the same measure in table 1, one does notice the big difference between them.

Comments
=========
1. Section 2, page2: “As derived in the original paper…” references which paper (i.e. Kingma et al)?
2. VAE or beta-VAE model is not referenced (mentioned on page 5);
3. Authors do agree that the corpora used is not optimal for the adequate evaluation of the proposed model. It would be interesting to see the use of this approach on a more realistic data set;
4. Unclear sentence: “Compared with the VAE, latent codes where images with the same labels are clustered.”;
5. The authors claim that one of the results of this work is the possibility to generalize for unseen cases (zero-shot learning). It would be interesting to see the performance of this model compared to SOTA in CV in terms of the zero-shot learning task;
6. Figure 2 visualizes proposed embedding space (2D) but it shows VAE and beta-VAE models and omits to show PSE. VAE and beta-VAE are neither introduced nor referenced in text;
7. Table 1: mark best performing with bold. It does, however, outperform other evaluated models when using 20D embedding space;
8. Page 7: in text you mention generation accuracy and in Table 1 the same value is defined as Generation Correctness (%).

---

> ### Author Response · Authors · 2018-11-20
> **Response to R2[1/2]**
>
> We would like to thank the reviewer for their thoughtful review and for taking the time to understand the essence of our paper. In the following response, we will directly address the reviewer's questions and comments, and also state how these are addressed in the revised paper.
>
> Questions
> =========
>
> 1. How was the word2vec model trained? Did you use an existing pretrained model (e.g. available as download) or did you train the embedding model yourself?
>
> We used the standard pre-trained word2vec model provided by Google (https://code.google.com/archive/p/word2vec/) to convert words to vectors. Of course, these vectors are then further processed by our encoder, so there is the possibility for our model to learn to change this embedding. We've made changes to the paper to make this clearer, and in response to other comments, we've also expanded the text to make our model variations much clearer.
>
> 2. The major novelty of this approach is the use of annotations supporting images and textual (pretrained) embedding spaces, but no related work regarding Wes was neither introduced in the Related Work section nor was it clearly explained in the text.
>
> We believe this has been addressed in the revised text. We would be happy to make any further changes if the reviewer feels this is necessary however.
>
> 3. Why did the authors focus on the w2v model instead of more promising approaches as fastText or ELMo?
>
> We believe that any word embedding model could be used, and only picked word2vec as an exemplar to demonstrate how the model responds to external information in terms of a pre-trained embedding. In future work, it would be interesting to explore how much of an effect different embeddings have, but to do this we would also want to seek richer images datasets to explore.
>
> 4. How does your model deal with OOV word(s) as input? For example, when used as Image Generator.
>
> If you mean OOV in terms of the image dataset labels, then this is covered in section 4.3 (of the version of the paper you reviewed), and in particular in figure 6, which shows images generated from words that are not part of the image label set. This is a key feature of our model. Using words that are out of vocabulary for the word embedding will however not produce sensible images.
>
> 5. Table 1 shows results achieved on MNIST but not Fashion-MNIST; was the evaluation performed on MNIST only?
>
> No, not at all, and this was only omitted to keep the paper length down. As you've highlighted this as something you'd like to see (and that we also wholeheartedly agree with), we've extended the presentation to include both MNIST and Fashion MNIST.
>
> 6. Table 2 presents the impact of the use of pretrained embeddings (word2vec) instead of one-hot vectors for labels. Which one do the models presented in Table 1 use?
>
> They use the one-hot variants. We believe our changes in the revised version of the paper to more clearly separate out the four variants (one-hot PSE, one-hot PSE*, w2v PSE, w2v PSE*) of the model that we experiment with.
>
> 7. Could you explain the small difference between using one-hot vs pretrained label encodings, presented in Table 2?
>
> The pre-trained word-vec embeddings impart additional contextual knowledge (the similarity of words), that we believe can subtly impose an additional prior on the layout of the probabilistic embedding space of our model. Somewhat surprisingly, it appears that this also has a small positive effect in terms of the objective performance measures on the datasets we experiment with. We note that in the original paper we did not quantify the significance of this effect, and this has been addressed in the revision.
>
> 8. Also, can you explain how the numbers in table 2 were achieved (e.g. sum over all, average of all, etc.). When comparing the values presented here to the values of the same measure in table 1, one does notice the big difference between them.
>
> Numbers in both tables are presented as averages over the test datasets. However, the datasets are different - Table 1 shows results on MNIST, whilst Table 2 on Fashion MNIST. Fashion MNIST has considerably more variability (and in particular many more non-black pixels) than MNIST. This should be much clearer in the revision that we have made.

---

> ### Author Response · Authors · 2018-11-20
> **Response to R2[2/2]**
>
> Comments
> =========
> 1. Section 2, page2: “As derived in the original paper…” references which paper (i.e. Kingma et al)?
> 2. VAE or beta-VAE model is not referenced (mentioned on page 5);
> 4. Unclear sentence: “Compared with the VAE, latent codes where images with the same labels are clustered.”;
>
> Thank you for spotting these - both points have been addressed in the revision.
>
> 3. Authors do agree that the corpora used is not optimal for the adequate evaluation of the proposed model. It would be interesting to see the use of this approach on a more realistic data set;
>
> We completely agree, however we do think this should be something in future work. Our aim with this paper was to demonstrate the power of the model over those previously proposed.
>
> 5. The authors claim that one of the results of this work is the possibility to generalize for unseen cases (zero-shot learning). It would be interesting to see the performance of this model compared to SOTA in CV in terms of the zero-shot learning task;
>
> Again, we completely agree, and that is a direction that we intend to study in depth going forwards. To be competitive on such a task, we would need to look at considerably more powerful encoder and decoder models however, and we feel that this would detract from the points we try to make in this paper.
>
> 6. Figure 2 visualizes proposed embedding space (2D) but it shows VAE and beta-VAE models and omits to show PSE. VAE and beta-VAE are neither introduced nor referenced in text;
>
> This is fixed in the revision - we have modified the figure to omit beta-vae (and replaced it with the missing PSE visualisation). The beta-VAE model was rather tangential to the discussion anyway as it is not conditioned on labels. The text has been modified accordingly and references added in the correct places.
>
> 7. Table 1: mark best performing with bold. It does, however, outperform other evaluated models when using 20D embedding space;
>
> This has been addressed in the revision.
>
> 8. Page 7: in text you mention generation accuracy and in Table 1 the same value is defined as Generation Correctness (%).
>
> Thank you for spotting this. We've moved to consistently using 'Generation Accuracy'.
>
> We hope that you are satisfied with the changes that have been made to the paper in light of your questions, comments and suggestions, and would like to thank you again for taking the time to help us improve our work. Please let us know if you have further questions or points you would like to see addressed.

---

### Public Comment · (anonymous) · 2018-11-26
**Model Structure**

Is the image encoder in PSE a three layer MLP with (28 * 28, 512) (512, 64) (64, latent_dimension)  or it's a two layer MLP with (28 * 28, 512) (512, latent_dimension)

---

> ### Author Response · Authors · 2018-11-28
> **To Model Structure**
>
> We use a three MLP with (28 * 28, 512) (512, 64) (64, latent_dimension).

---

### Meta-Review · Area_Chair1 · 2018-12-20
**limited novelty and limited experimental evaluation**

**Confidence:** 5
**Recommendation:** Reject

**Metareview:**

mnist and small picture variants are not that impressive.
it is a minor extension of VAEs which also are not common in sota systems.